# Secretome Analysis of the Plant Biostimulant Bacteria Strains *Bacillus subtilis* (EB2004S) and *Lactobacillus helveticus* (EL2006H) in Response to pH Changes

**DOI:** 10.3390/ijms232315144

**Published:** 2022-12-02

**Authors:** Levini A. Msimbira, Sowmyalakshmi Subramanian, Judith Naamala, Mohammed Antar, Donald L. Smith

**Affiliations:** Department of Plant Science, McGill University, Montreal, QC H9X3V9, Canada

**Keywords:** pH, secretome, proteomics, plant microbe interaction, biostimulant, plant growth promotion

## Abstract

It is well-known that there is a high frequency of plant-growth-promoting strains in *Bacillus subtilis* and that these can be effective under both stressful and stress-free conditions. There are very few studies of this activity in the case of *Lactobacillus helveticus*. In this study, the effects of pH on the secretome (proteins) in the cell-free supernatants of two bacterial strains were evaluated. The bacteria were cultured at pH 5, 7 and 8, and their secretome profiles were analyzed, with pH 7 (optimal growth pH) considered as the “control”. The results showed that acidity (lower pH 5) diminishes the detectable production of most of the secretome proteins, whereas alkalinity (higher pH 8) increases the detectable protein production. At pH 5, five (5) new proteins were produced by *L. helveticus*, including class A sortase, fucose-binding lectin II, MucBP-domain-containing protein, SLAP-domain-containing protein and hypothetical protein LHEJCM1006_11110, whereas for *B. subtilis*, four (4) types of proteins were uniquely produced (*p* ≤ 0.05), including helicase-exonuclease AddAB subunit AddB, 5-methyltetrahydropteroyltriglutamate-homocysteine S-methyltransferase, a cluster of ABC-F family ATP-binding-cassette-domain-containing proteins and a cluster of excinuclease ABC (subunit B). At pH 8, *Bacillus subtilis* produced 56 unique proteins. Many of the detected proteins were involved in metabolic processes, whereas the others had unknown functions. The unique and new proteins with known and unknown functions suggest potential the acclimatization of the microbes to pH stress.

## 1. Introduction

Plants account for more than 90% of the biomaterial in the global food chain, and their coexistence with microbes has important evolutionary significance. Recently, research on microbe-based biostimulants has taken a major leap, as low-cost and environmentally friendly agricultural inputs, and a specific set of plant–microbe interactions are now seen as forming the holobiont [1,2,3]. Comprehensive studies of the whole protein profiles of organisms, tissues or cells are now more precisely studied using various omics platforms [4,5,6].

Plant-growth-promoting bacteria (PGPB) have been recognized for their contributions to agriculture and environmental sustainability for more than a century [7]. *Bacillus subtilis* is one of the most widely studied groups of bacilli, representing a large group of rhizosphere prokaryotes. *B. subtilis* is known for biofilm formation in the rhizosphere during root colonization, facilitating nutrient uptake by plants [8]. The ability to produce endospores makes *B. subtilis* comparatively successful in terms of survival and plant colonization relative to non-spore-forming PGPB [9]. Lactobacilli are a group of lactic acid bacteria (LAB) that make a valuable contribution to soil organic matter production, improving soil health and fertility. LAB have been recognized for their probiotic benefits in fermented food products, but their importance has now been extended to sustainable agriculture applications [10]. The expansion of knowledge in this area has followed the continuum of challenges and limitations involved in understanding the roles of these microbes under adverse environmental conditions and how they can aid in plant stress alleviation. Their potential for a consistent performance has often diminished as experiments moved from controlled laboratory conditions to greenhouses and then to the open field [11]. Because different microbes use different systems to sense and respond to environmental cues, studying how they behave under normal and stressed conditions makes it easier to understand their survival and, consequently, their potential to promote plant survival and development.

Like other unicellular organisms, microbes struggle to deal with the deleterious effects of varied levels of proton concentrations (hydrogen ions (H^+^), measured as pH). The PGPB surrounding plant roots are affected by both the root exudes and available soil nutrients. For example, when ammonium is a nitrogen source, additional H^+^ is released, which reduces the pH and may be detrimental to both the plants and their associated microbes [12,13]. Given the often narrow pH range that allows for the vigorous growth of soil microbes, soil pH fluctuation is among the most frequent environmental stressors affecting microbes [14], including those associated with domesticated plants. There are even concerns about this process in regard to marine habitats, where increased CO_2_ emissions have impacted the chemistry of oceans, leading to decreased pH levels [15]. To avoid the direct effects of otherwise stressful pH conditions on crop plants, we applied a microbial cell-free supernatant (CFS) to enhance plant growth and worked to isolate and identify microbially produced growth-promoting/bioactive compounds [16,17,18,19].

The considerable increase in the use of proteomic tools and technologies has led to an improved understanding of the underlying mechanisms of the interactions between microbes and their environment. Our knowledge of proteomics has considerably facilitated the study of the contributions of microbes to plant stress tolerance and growth promotion [16,20,21]. However, microbial CFS proteomic analysis information remains fragmented [16,22], especially for PGPB under acid and alkaline stress conditions. CFSs are known to contain important compounds [23], the bioactivity of which could have profound effects on plant defenses against stress and could be applicable to plant growth stimulation and improved crop production. Studies of CFS aiming to determine the antimicrobial compounds secreted by the *Bacillus thuringiensis* strain NEB17 [24] and *Pseudomonas entomophila* 23S [25] resulted in the identification of bioactive products that have since been moved toward commercialization. However, secretome profiling based on the CFS of microbes cultured at varying pH levels is lacking, and knowledge of this area could enable the discovery of more positive effects on plant growth promotion.

The current study is focused on investigating the effects of pH and the mechanisms of tolerance in two bacterial strains (*Bacillus subtilis* (EB2004S) and *Lactobacillus helveticus* (EL2006H)), which are part of the five individual proprietary microbial strains constituting a synthetic microbial consortium from EVL Inc. (http://www.evlbiotec.com, accessed on 17 July 2022), currently used as a plant biostimulant technology. Because pH has been used as a model parameter in studies of the microbial species composition of soils [26] and in the human gut [27], the aim of this study was to examine, at pH 5, 7 and 8, the exoproteome profiles of two microbial strains with CFSs that cause plant growth promotion [18]. Because microbial growth is known to strongly modify the pH of the environment, affecting microbial interactions with plants [28], in this study, we attempted to better understand the strain behavior and tolerance to pH as a prerequisite for understanding their growth profiles, as a quantitative measure of the difference between stressed and unstressed secretomes. The protein profiles were analyzed during the late exponential growth stage (48 h) of the cultures of both *L. helveticus* (EL2006H) and *B. subtilis* (EB2004S).

## 2. Results

Most organisms are only capable of surviving within a relatively narrow pH range. As a result, this is regarded as one of the cardinal parameters of life. Limits on the tolerance to pH variation represent one of the basic properties of most microbes, including *B. subtilis* [29] and *L. helveticus* [30]. The use of microbes and microbial products as biofertilizers has dramatically increased in recent years. In most cases, these microbes are used and/or produced under optimal conditions before they are applied. In the current study, we focused on building on the previous study regarding the use of the cell-free supernatants of *B. subtilis* (EB200S) and *L. helveticus* (EL2006H) [18]. Given the positive results, we focused on understanding the secretome protein profiles as a mechanism of adaptation to the varying pH levels of the growth media while benefiting the plant. To achieve this, the proteome profiles of the *B. subtilis* (EB2004S) and *L. helveticus* (EL2006H) CFSs, which have positive effects on the seed germination and seedling growth of corn and tomato [18], were analyzed using LC/MS spectrometry, and the results were as indicated below.

### 2.1. Proteome Analysis of L. helveticus (EL2006H) CFS

Scaffold and OmicsBox analyses of the LC-MS data resulted in the deferential identification and quantification of the proteins of the *L. helveticus* CFS when the strain was cultured at pH 5 and 7 (Table 1, Appendix A). There was substantial variations in the total number proteins identified for both pH levels. A total of 176 proteins were detected for pH 5 and 553 were detected for pH 7 in *L. helveticus* CFS, among which 167 proteins identified were found in the case of both pH level treatments. The first visualization was based on Venn diagrams, which summarized the total unique spectra, total unique peptides and total unique proteins identified from the samples within the set threshold levels. In this study, the Venn diagrams (Figure 1) were generated based on threshold values of ≥2 peptides and a false discovery rate (FDR) of ≤0.01, which provided the first indication of the degree to which predictor variables of the MS spectra and peptides of the proteins were within the threshold values for *L. helveticus* (Figure 1). Venn diagrams have been used to categorically illustrate the relationships between proteins, total unique peptides, and total unique spectra [31], which is a key pre-requisite for the selection of the best variables based on the interests of the study.

There was an approximately 42-fold decrease in the number of unique identified proteins at pH 5, which indicated a strong effect of acidity on the strain (Figure 1). With reference to pH 7 as the optimum pH for the consortium, when produced commercially in a bioreactor, *L*. *helveticus* had approximately 3-fold more secretome proteins than the cultures at pH 5. Observations regarding the two pHs tested for the total proteins, unique peptides and spectral quality between the acidic pH 5 and reference pH 7 are depicted in Figure 1. Since *L. helveticus* CFS did not show any significant plant growth promotion at pH 8, as reported previously [18], this level of pH was not studied further for this strain.

### 2.2. Groups of Identified Proteins from L. helveticus (EL2006H) CFS and Their Functional Annotation

The differentially identified proteins (DIPs) were studied by gene ontology (GO) enrichment analysis, which generated four major functional clusters based on the cellular components, biological processes, molecular functions and enzyme code distribution (Figure 2).

Under the biological process functional cluster, the most frequently identified proteins were those related to organic substance metabolic processes and primary metabolic processes at pH 7, with an ~10-fold increase, as compared to those identified at pH 5 (Figure 2). Proteins related to small-molecule metabolic processes and the establishment of localization were identified only at pH 7 (Figure 2). Proteins related to cell division, protein folding, cell wall organization or biosynthesis, anatomical structure morphogenesis, anatomical structure development, the regulation of biological quality, transmembrane transport and the regulation of developmental processes were identified only at pH 5 (Figure 2).

Based on the molecular function, protein binding, catalytic activity acting on the protein and hydrolase activity were detected at pH 5, while most proteins present at pH 7 were those with organic cyclic compound binding and heterocyclic compound binding activities (Figure 2), suggesting that pH variation has a substantial influence on bacterial physiology.

The enzyme code distribution was also greatly influenced by the pH. A lower identification was detected at pH 5 than pH 7 (Figure 2 and Table 2). At pH 5, a decrease to zero activity was observed for some enzymes. This corresponded to the lower overall activity of the cells, reduced molecular function and diminished biological processing at pH 5 [18].

The cell components also manifested substantial variation in terms of the decrease in proteins produced as a result of the pH change. The proteins responsible for membrane production and the intrinsic components of membranes were more upregulated at pH 7 than pH 5 (Figure 2).

### 2.3. Identified Proteins from L. helveticus (EL2006H) CFS

The further quantitative analysis of the LC-MS output resulted in the identification of proteins identified in response to pH variation. A comparison of the CFS proteins using the Fisher exact test (Figure 2 and Figure 3, Appendix A) showed significant decreases in the proteins at pH 5 when compared with pH 7. Five proteins, including class A sortase, fucose-binding lectin II, MucBP-domain-containing protein, SLAP-domain-containing protein and hypothetical protein LHEJCM1006_11110, were uniquely (*p* ≤ 0.05) derived from the pH 5 CFS. About 54 proteins were upregulated at pH 5, with reference to pH 7, while most of the proteins were downregulated, and some were not detected at all at pH 5 (Appendix A). Fold change analysis was also used to quantify the unique proteins in this study. A cut-off of a ±1.5-fold change was regarded as statistically significant for functional interpretation (Figure 3).

### 2.4. Proteome Analysis of B. subtilis (EB2004S) CFS

Scaffold and Blast2GoPro analyses of the LC-MS/MS data resulted in the deferential identification and quantification of the proteins of the *B. subtilis* (EB2004S) CFSs when the strain was cultured under varying pH conditions (Figure 4, Appendix A). The trend observed across the pH range tested demonstrated that the total proteins, unique peptides and spectral count decreased at the acidic pH of 5 and increased at the alkaline pH of 8, as compared to the reference pH 7 (Figure 4).

### 2.5. Groups of Identified Proteins from B. subtilis (EB2004S) CFS and Their Functional Annotation

The GO enrichment analysis was conducted to identify DEPs and generate major functional clusters based on the cellular components, biological processes and molecular functions (Figure 5).

In the biological process cluster, the largest number of identified proteins were found at pH 8, and these were related to organic substance metabolic processes, cellular metabolic processes, primary metabolic processes, nitrogen compound metabolic processes, biosynthesis processes and cellular metabolic processes. At pH 5, the identification of unique proteins related to the response to stress was evident. GO terms for cell division, cell wall organization or biosynthesis, anatomical structure morphogenesis, anatomical structure development, the regulation of biological quality, transmembrane transport and the regulation of developmental processes, cellular homeostasis, the response to external stimuli and NADH metabolic processes were present only at pH 5 (Figure 5).

Based on the molecular function, pH 5 resulted in the unique identification of four proteins, including those for sulfur compound binding, peroxiredoxin (WP_003236447.1) activity, amide binding and peroxidase activity, in comparison with pH 7 and 8 (Figure 4). Lyase activity was uniquely identified only at pH 7. Many of the clusters detected at pH 7 and 8 were for ion binding, organic cyclic compound binding, heterocyclic compound binding, small molecule binding and hydrolase activity. There were similar trends in the profiles at both these pHs (Figure 5).

The enzyme code distribution was also greatly influenced by the pH. Fewer identifications were determined at pH 5 than pH 7 and 8 (Figure 5 and Table 2). At pH 5, there was a significant decrease to zero activity for some enzymes. This corresponded to the activity of the cells, molecular function and biological processes, which were lower at the lower pH of 5. More specifically, the ligases and translocases did not express their activity at pH 5, suggesting that the bacteria were attempting to modulate their enzymatic profiles in order to accommodate normal functioning under the acidic pH 5. The pH changes also affected the cell components and proteins responsible for intracellular anatomical structure, cytoplasm activity and membrane production. These were upregulated at pH 7 and 8 and downregulated at pH 5 (Figure 5).

### 2.6. Identified Proteins with Differential Identities from B. subtilis (EB2004S) CFS

The quantitative analysis of the LC-MS results for the *B. subtilis* CFS led to the significant identification of proteins produced in response to the pH variation. Our comparison of the CFS proteins using the Fisher exact test (Figure 6A,B, Appendix A) showed significant decreases in the identified proteins. However, four significantly unique sets of proteins, including helicase-exonuclease AddAB subunit AddB, 5-methyltetrahydropteroyltriglutamate-homocysteine S-methyltransferase, a cluster of ABC-F family ATP-binding-cassette-domain-containing proteins and a cluster of excinuclease ABC (subunit B), were identified at pH 5. Chain B, selenosubtilisin (pdb|1SEL|B), was the only protein upregulated at pH 5, with reference to pH 7, while the majority of the proteins were downregulated, with some not being detected at all (Figure 6A, Appendix A). The comparison of the CFS proteome at pH 7 and 8 showed an increased number of proteins, including both upregulated proteins such as isochorismate synthase DhbC (WP_060399405.1 [3]) and ABC-F family ATP-binding-cassette-domain-containing protein (WP_003239940.1), and downregulated proteins, such as beta-mannosidase (WP_060399088.1) and right-handed parallel beta-helix repeat-containing protein (WP_004429655.1) (Figure 6B).

## 3. Discussion

Microbes are regarded as the most successful life forms, having survived sometimes extreme environmental changes and having developed a wide range of adaptation mechanisms in response to changing habitat conditions. Like other cells, microbes struggle to deal with the deleterious effects of varying levels of proton concentrations. Most microbes have developed various secretory mechanisms which help them to interact amongst themselves and with their hosts [35], many of which are not yet understood. The two strains studied here, *B. subtilis* and *L. helveticus,* are both Gram-positive. The *Bacillus* and *Lactobacillus* types are widely used in the food industry, particularly in probiotics and fermentation processes, as well as plant growth biostimulant products. This study of the CFS proteomes for *B. subtilis* (EB2004S) and *L. helveticus* (EL2006H) focused on understanding the pH variation adaptation mechanisms and possible similarities and dissimilarities which could help us in exploring their potential to provide pH change benefits. As biostimulant strains, the output of their secretome could also be related to plant growth stimulation.

The study revealed that the bacterial CFS proteome, when compared with the neutral pH 7, was affected by both lower (acidity, pH 5) and higher (alkalinity, pH 8) proton concentrations. Upregulation and downregulation in protein identification, because of stress tolerance, are known phenomena of bacteria [36,37], even though there are differences in the specific levels of identification [38].

Microbes develop various adaptive mechanisms when they are exposed to stressors such as more extreme levels of salinity, temperature and/or pH. The mechanisms range from genetic and cellular to membrane modulations. *Bacillus subtilis* is one of the most widely studied bacteria and is a powerful genetic resource, as well as a model for cell development and differentiation [39,40]. It is of growing commercial importance due to its ability to secrete proteins and low molecular compounds, and it is also a well-known plant-growth-promoting bacteria [23]. Despite the lack of an outer membrane in *B. subtilis* [41], the bacterium exhibits a great potential for the production of detected cytoplasmic proteins and specific secretion abilities [35,42,43]. In most cases, the detection of cytoplasmic proteins in the growth medium is associated with cell lysis [44,45]. However, protein presence within extracellular media without cell lysis has also been reported in studies of bacteria [46]. This type of protein secretion is termed as non-classical [44,47], which, indeed, can be seen in the current study in the differentially identified proteins, which were quantified in the CFSs of *B. subtilis* and *L. helveticus*. In *L. helveticus*, for instance, the significant upregulation of a membrane protein (GFP02734.1 (+2)) with a smaller molecular weight of 15 kDa was evident at *p* ≤ 0.05. This protein, together with S-layer protein precursors and SLAP-domain-containing proteins, might have originated from cell lysis due to the acidity at pH 5. In earlier studies, these SLAP proteins of *Lactobacillus* were only known to function in terms of adhesion to the intestinal cells [48,49,50] and host immunity modulation [51,52]. Knowledge regarding their functional spectrum has now increased with respect to cell maintenance activities, such as their cell division [53] and cell autolysin roles [54]. The upregulation of these proteins in *L. helveticus* CFS at pH 5 suggests that the proteins function to maintain cell function under acidity stress.

On the contrary, *B. subtilis* has many secretory pathways, which, in most cases, leave the cell intact, but large quantities of proteins are secreted into the medium [55]. The sec-dependent secretory pathway is responsible for the majority of proteins exported from the cytoplasm to the growth medium, which typically contain cleavable N-terminal signal peptides [56,57]. Other proteins are secreted through twin-arginine translocation (TAT) systems [58]. In addition to the known pathways for protein secretion in *B. subtilis*, alternative mechanisms of the release of signal-less proteins into the growth medium are known [59,60]. This makes it difficult to determine whether the studied proteins in the current research are the result of cell lysis or not. Previous studies also showed ambiguity in determining whether the proteins of the CFSs are directly derived from cell lysis, especially for *B. subtilis* [61,62]. Similar to *B. subtilis*, non-classical secretion has also been reported in the case of *L*. *crispatus*, which secretes the pH-dependent proteins glyceraldehyde-3-phosphate dehydrogenase and enolase only under alkaline conditions [63,64]. Uniquely, in this study, the two mentioned proteins were not secreted at pH 5 in the case of *L. helveticus* but were highly upregulated at pH 7, which further explains the presence of these proteins in the media. They are pH driven, as the pH moves towards alkalinity.

At the cellular level, *B. subtilis* combats stressful conditions by producing stress-resistant endospores, while at the microbial community level, it can take up external DNA as a means of adaptation by recombination. However, a quick response to acidity or alkalinity is achieved through the activation of alternative sigB, which is a known regulator of the stress response [65,66]. Specifically, under higher proton (acidic stress) concentrations, *B. subtilis* activates tolerance mechanisms such as proton-consuming decarboxylase reactions, the pumping out of protons, causing lipid content modifications, and ammonia production [67,68]. Generally, when a cell is exposed to acidic conditions below its optimum pH, the proteins become damaged and accumulate in the cytoplasm. One piece of evidence suggesting a low pH effect on the cells is decreased enzyme activity in the cytoplasm, a clear signal that the optimum pH range has been exceeded for metabolic enzymes. The decrease in the enzyme code distribution found in the *B. subtilis* CFSs in the current study substantiated the observation that the metabolic activities at pH 5 were decreased (Table 2). A decrease in the hydrolase enzyme class representation by 94% (Table 2), as reported here, suggested that some of the enzymes, such as the proteases, which are normally extracellular with a good stability, were not functional [69,70]. Furthermore, bacteria of the *Bacillus* genus are known for their commercial production of proteases, especially neutral and alkaline ones [71], which, again, is reflected in the current study in the greater representation of the hydrolase enzyme class at both pH 7 and 8. Some microbes induce components of the Clp protease complex to participate in protein homeostasis by removing the damaged/degraded proteins. Conversely, depending on the microbial species, periplasmic (HdeA and HdeB) and cytoplasmic (DnaK and GroEl) chaperons are induced in response to acid stress [72,73].

Understanding the pH variation responses of most microbes is an ongoing area of investigation, especially for neutrophilic bacteria such as *B. subtilis*. The maintenance of intracellular pH homeostasis is achieved through proton motive forces created by respiration around the cytoplasmic membrane, which works simultaneously with the extrusion of toxic Na+ ions from the cell while it takes in protons. The Na^+^/K^+^/H^+^ antiporter, a major system of pH homeostasis, is encoded by the tetA(L) gene in *B. subtilis* [74,75,76,77,78]. At an elevated pH of about 8, it was reported that the tetA(L) gene was upregulated by 1.8–2.5 fold [76].

Some of the universal stress proteins responsible for alkalinity stress tolerance, such as ClpB, belong to the Clp family of ATPases, containing chaperone proteins responsible for stabilization and aiding in the correct folding and assembly of denatured proteins [79,80]. Generally, these proteins minimize the accumulation of improperly folded and damaged proteins, and the current study measured the significantly increased presence of these proteins at pH 8 (Figure 6B).

Chain B, selenosubtilisin (pdb|1SEL|B) protein, which was the only upregulated protein in this study at pH 5 in the case of *B. subtilis* CFS (Figure 6A), is one of the better-studied acyl transferases [81]. Industrial selenosubtilisin is exploited for its capacity for amide hydrolysis, although it is a poor catalyst compared to the protease subtilisin from bacteria. Selenosubtilisin is also a catalyst of alkyl hydroperoxide reduction by thiols [82,83], an action mirroring that of glutathione peroxidase, an important mammalian enzyme. Naturally occurring selenoproteins are involved in cellular redox reactions. These reactions have important roles in the detoxification of organic hydroperoxide and the protection of membrane lipids from oxidative damage [84]. Because *B. subtilis* is a neutrophilic bacterium, the upregulation of Chain B, selenosubtilisin, suggests the protection against cell membrane damage resulting from the high proton concentration in the growth medium.

A group of ABC-F family ATP-binding-cassette-domain-containing proteins were also identified in the pH 5 CFS proteome as being among the unique proteins produced by *B. subtilis*. This super family of proteins, or traffic ATPases, are responsible for the translocation of solutes across the membrane [85]. Their function is not limited to nutrient acquisition but is also involved in signal transduction, bacterial pathogenesis and protein secretion [86,87]. Whether or not the unique exoprotein identified in the current study was functionally associated with the low pH tolerance requires further investigation. DNA repair and maintenance are very important for cells, especially when the cells attempt to adapt to stressful environments. A protein from the cluster of excinuclease ABC (subunit B) proteins was uniquely identified in the CFS of *B. subtilis* at pH 5, a protein that is known for DNA damage repair [88]. Due to pH disruption of normal function DNA repair proteins, alternate repair proteins are necessary to maintain cell functionality.

The *L. helveticus* studied belongs to the lactic acid bacteria (LAB), which are so named because they produce lactic acid as a major product of their metabolism [30]. LAB are among the major groups comprising the human microbiome and have applications in the food industry and as probiotics. *Lactobacilli* have also been utilized as PGPR and enhance plant growth through the production of auxins, plantaricin and volatile fatty acids, in addition to providing protection against phytopathogens [89,90,91]. The *L. helveticus* (EL2006H) used in this study is a member of a microbial consortium that is effective as a plant growth biostimulant, which showed a positive seed germination bioactivity [18]. Most LAB are known to produce acid, which in itself lowers the pH of the habitat, or the culture medium in this case. This group of Gram-positive bacteria are usually under the constant pressure of exposure to low pH, which has caused them to develop various tolerance mechanisms. In the human gut, for example, microbiota such as *Lactobacilli* must adapt to low pH in order to be functional as they move through the gastrointestinal tract (GIT) [92]. To deal with low-pH stress, LAB produce alkaline substances such as ammonia, urea and arginine to neutralize acidic conditions [93]. The arginine dihydrolase system (ADS) aids in catalyzing the conversion of arginine into ammonia, ornithine and carbon dioxide (CO_2_), while urea is hydrolyzed by ureases into ammonia and CO_2_ [93]. Similarly, malolactic fermentation by LAB liberates CO_2_, which also neutralizes elevated proton concentrations [94].

A class A sortase protein was found to be produced at pH 5 by *L. helveticus*. Previous studies have reported this protein as a surface-anchoring protein [95]. This protein has also been identified for its ability to anchor *L. acidophilus*, allowing it to persist in the GIT [96], which is a low-pH environment. Probably, this protein is unique in its ability to both persist under low pH conditions and help to anchor *L. helveticus* in low-pH environments. The adhesion adaptation mechanisms of the MucBP-domain-containing protein, involved in the retention of LAB in the GIT, are already known [97]. The MucBP-domain-containing protein was uniquely identified in the *L. helveticus* CFS at pH 5. This might be the first time that this type of protein was associated with pH tolerance mechanisms, in addition to its already known adhesion function. The SLAP-domain-containing proteins were also identified to be unique to pH 5. Previous studies have reported the presence of these proteins in the *L. acidophilus* group of *Lactobacilli*, which includes *L. helveticus* [98,99,100]. SLAP-domain-containing proteins are another group of exoproteins involved in intestinal adhesion, as in the case of *L. helveticus* [101] and *L. amylovorus* [102]. The current study lacks explicit confirmation regarding the function of these proteins, specifically in regard to the pH tolerance of the strain, but it does indicate that the strain could probably adhere itself to a host organism, even under acidic conditions, or to other members of its species in order to form protective biofilms.

On the other hand, LAB produce lectins, which help to maintain relationships between the host and microbes. They are determinants of symbiotic tropism or pathogenicity during the interaction with glycans [103]. Lectins are important for cell aggregation, showing potential for application in the areas of bioremediation and antimicrobial activity. A lectin of 30 kDa was reported to have antimicrobial activity against Gram-positive and Gram-negative bacteria [104]. The current study, by contrast, identified a 37 kDa fucose-binding lectin II (Appendix A), which might have been activated in response to the pH as a tolerance mechanism at pH 5 and could serve as an effective plant biostimulant (adhesion and symbiosis [105]), in addition to its stress tolerance function.

C39 peptidase (WP_172994769.1) is a family of proteins involved in bacteriocin processing endopeptidases. These proteins cleave N-terminal leader peptides synthesized in the precursor stages of different bacteriocins at the maturation stage [106,107]. The mediators of the cleavage of the leader peptides are membrane proteins, which take part in secretion processes. The fact that this family of proteins was upregulated is consistent with earlier studies, showing that the activity of cysteine peptidases is related to acidic pH conditions and that proteins with this activity are present only in acidic conditions, such as those in the plant vacuole and animal lysosomes [107]. The increased presence of bacteriocins means that increased pathogen resistance would indirectly be conferred on the plant, adding to growth stimulation or plant health, as many pathogenic bacteria are inhibited.

PASTA-domain-containing proteins have previously been identified as human pathogens, such as *Staphylococcus aureus*, which are involved in cell wall metabolism, increasing pathogen virulence and causing resistance to drugs [108,109]. These proteins are further known as sensor motifs for the binding of beta-lactam compounds and fragments of cell walls, such as peptidoglycans [110]. The current study revealed the downregulation of the Stk1 family PASTA-domain-containing Ser/Thr kinase—QPB51502.1 (+1) (Figure 3). This suggests that the cell wall was impacted by the lower pH of the growth medium. Cold-shock protein (QPB52189.1) and a cluster of extracellular solute-binding (QPB52160.1) proteins were among the other downregulated proteins. Cold-shock proteins are usually known to be produced in response to rapid temperature decline [111] and certain other abiotic stresses [112]. Interestingly, these proteins are known to decrease their synthesis immediately when the cells encounter lower temperatures, enabling increases in the synthesis of other proteins [111]. The downregulation of the cold-shock stress proteins observed in this study could be an emulation of this response.

Proteins of unknown function, such as hypothetical protein LHEJCM1006_11110, were uniquely identified at pH 5 in the current study, while hypothetical protein DQL94_04110 (AZA21465.1) and hypothetical protein LH5_01156 (AZK91398.1), among others, were upregulated at pH 5 (Figure 3). A protein remains hypothetical or putative in the database because there is a corresponding mRNA sequence available, but there are no other experimental data to show that they exist. Such proteins, which lack sufficient functional annotation, are commonly identified with respect to the untargeted protein profiling of microbes under both optimum and stressed conditions. Proteins of unknown function from the secretome of *Bacillus lehensis* were identified, as reported earlier [22], and four hypothetical proteins were identified, bearing the accession numbers AIC96118, AIC94431, AIC96385 and AIC96376. A study by the authors of [113] focused on the reported presence of other hypothetical proteins in *Exiguobacterium antarcticum*. Such proteins, given the conditions in which they were identified, might be produced because of a particular stimulus involved in the experiment, which could help researchers to predict their functions. However, the lack of matching sequences for the proteins of known function among the identified hypothetical proteins is a challenge. For example, [114] identified pH-related hypothetical proteins in cyanobacteria. This study, to the best of our knowledge, was the first report on CFS secretome analysis in the context of pH variation and plant growth promotion. Thus far, it is not clear whether the observed positive bioactivity of the *B. subtilis* and *L. helveticus* CFSs previously observed [18] are directly linked to the identified proteins or other biomolecules. Meaningful insights were acquired, as significantly unique proteins were identified, and others were found to be upregulated as a result of the pH treatments.

## 4. Materials and Methods

### 4.1. Microbial Strain Growth Conditions and Media

The possible survival mechanism(s) of individual strains of the consortium were assessed through the proteome profile analysis of the CFSs of *B. subtilis* (EB2004S) and *L. helveticus* (EL2006H), following previously described culture and growth conditions (Msimbira, Naamala, Antar, Subramanian and Smith [18].

#### Proteomics Study from the CFSs

To examine the secreted proteins resulting from the variation in the pH of the culture medium, the late exponential growth phase cultures (48 h) of the strains were centrifuged in nutrient broth (NB) at 4 °C and 10,000 rpm (15,180× *g*; SLA-1500) on a Sorvall Biofuge Pico (Mandel Scientific, Guelph, ON, Canada) for 10 min to separate the bacterial cell-free supernatant from the bacterial cells. A further purification process was conducted through vacuum filtration using a 0.22 μm filter to ensure that no cells remained in the CFS. The total proteins were extracted from the CFS using trichloroacetic acid (TCA; T9151, Sigma Aldrich) precipitation. Then, 100% (*w/v*) TCA was mixed with the bacterial supernatant to create a 25% concentration. The solution was then mixed, kept at −20 °C for 1 h, and then shifted to MBI orbital shaker at −4 °C with a shaking speed of 90 rpm (Montreal Biotech Inc., Montreal, Canada) overnight to enable protein precipitation. The precipitated material was again centrifuged at 4 °C for 10 min at 10,000 rpm, and the supernatant was discarded to obtain a protein pellet. The protein pellet obtained was washed several times with ice-cold acetone, air-dried under a laminar flow hood and dissolved in 2 M urea (U4883, Sigma Aldrich Canada Co., Ontario, Canada). Each TCA precipitation experimental treatment was replicated four times, and the whole experiment was repeated three times independently. To obtain sufficient protein samples, the four replicates per treatment were pooled together to form one sample. The protein concentration assay was determined following the method of Lowry, et al. [115].

### 4.2. Protein Profiling

A liquid chromatography mass spectrometry (LC-MS/MS) approach was used for the CFS protein profiling analysis, and the samples were analyzed by LC-MS/MS at the Montreal Clinical Research Institute (IRCM). A minimum of 10 μg of protein was dissolved in 20 μL of 2 M urea and then sent for analysis. The total proteins were digested with trypsin and injected into an LC-MS/MS equipped with a Linear Trap Quadrupole Velos Orbitrap (Thermo Fisher, Waltham, MA, USA). The subsequent analysis for the protein identification was conducted following established methods [16,116]. The dataset obtained from the mass spectra were searched against *Lactobacillus* spp. and *Bacillus* spp. databases, respectively, using Mascot software (Matrix Science, London, UK).

### 4.3. Protein Identification Confidence Level

The validation of the MS/MS-based peptides and proteins identified was conducted using Scaffold Software (version 5.1.2, Proteome Software Inc., Portland Oregon, USA). A probability greater than 95% was used to accept any identified peptide, as specified by Keller, et al. [117]. Similarly, protein identification at a probability greater than 95% was accepted as assigned and as having at least 2 identified peptides. Scaffold was used for the quantification of the proteins based on their spectra count values. These spectral count values were normalized, and analysis of variance (ANOVA) was performed to detect the differential abundance between the treatments at the 95% confidence level (*p* < 0.05).

### 4.4. Data Analysis

The proteomic data for the identified proteins obtained from the LC-MS/MS analysis were quantitatively analyzed for the fold change using Fisher’s exact test with Scaffold 5 (Scaffold Software for MS/MS Proteomics) to determine the significant differences. Data normalization was achieved by using threshold values for the protein and spectra while ensuring that the minimum number of peptides was 2. These spectral count values were normalized, and analysis of variance (ANOVA) was performed to detect differential abundance between the treatments at the 95% confidence level (*p* < 0.05) using the Benjamini–Hochberg multiple test correction. Analyses of the FASTA files generated from Scaffold 5 were performed using OmicsBox for the functional annotation and interpretation of the protein sequences. The LC-MS/MS proteomic data are available in the Mass Spectrometry Interactive Virtual Environment (MassIVE) under the dataset identifiers PXD036602 or doi:10.25345/C5XD0R27D for *Bacillus subtilis* and PXD036598 or doi:10.25345/C5PV6BB9K for *Lactobacillus helveticus*.

## 5. Conclusions

In agriculture and the food industry, the acid and alkalinity tolerance of bacteria has gained importance. In probiotics, for instance, bacteria well adapted to low pH conditions are preferred, as they contribute good levels of nutraceuticals to the food and, hence, improve gastrointestinal health through the secretion of useful functional molecules [118]. The same applies to microbes in the context of agriculture and climate change, where acidity and alkalinity are effectively alternate terms for the wet and arid regions in the world, respectively. The bacteria studied here are components of a multi-genus consortium that we are trying to understand as individual bacteria. Both the *B. subtilis* (EB2004S) and *L. helveticus* (EL2006H) CFS proteome analyses showed a wide range of pH adaptation through differential protein expression. The proteomics approach helped us to identify new unique proteins at pH 5 (five for *L. helveticus* and four for *B. subtilis*), whose functions are not fully understood. The study also revealed that when *B. subtilis* (EB2004S) and *L. helveticus* (EL2006H) were subjected to pH levels beyond their optimum range, significantly unique proteins were identified, which will increase interest in further investigations regarding the ways in which these new proteins help the bacteria to manage stress and, possibly, plants under similar conditions.

## Figures and Tables

**Figure 1 ijms-23-15144-f001:**
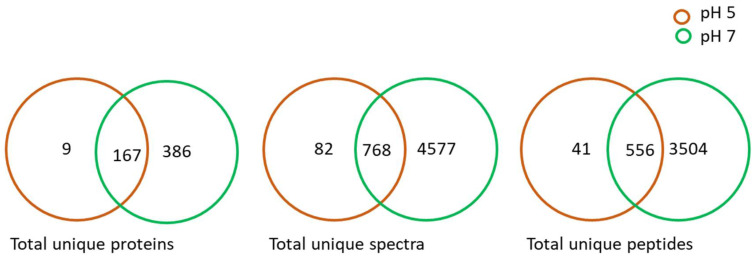
Venn diagram summarizing the total unique proteins, spectra and peptides obtained from *L. helveticus* (EL2006H) CFS at pH 5 and pH 7.

**Figure 2 ijms-23-15144-f002:**
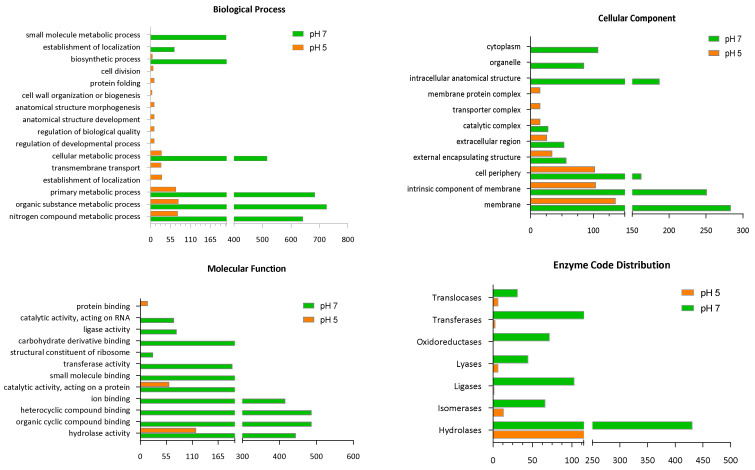
Variation in biological processes, cellular components, molecular function, and enzyme code distribution of the proteins based on the GO enrichment analysis of *L. helveticus* obtained from CFS when cultured at pH 5 and 7.

**Figure 3 ijms-23-15144-f003:**
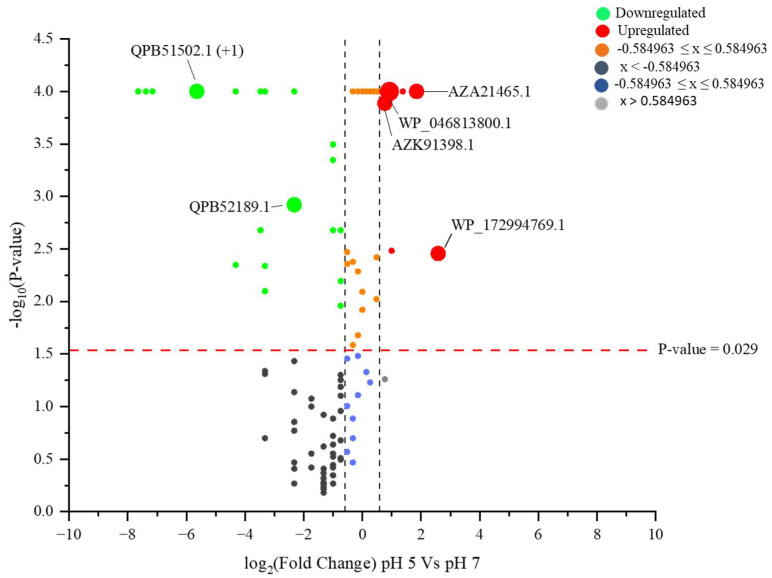
Volcano plot illustrating the distribution of differentially quantified proteins from *L. helveticus* CFS as −log10 (Benjamini–Hochberg-adjusted *p*-values) plotted against log2 (fold change) for pH 7 vs. pH 5. The non-axial black dotted vertical lines represent a ±1.5-fold change, while the horizontal red dotted line indicates the significance threshold (before logarithmic transformation) at *p* ≤ 0.029.

**Figure 4 ijms-23-15144-f004:**
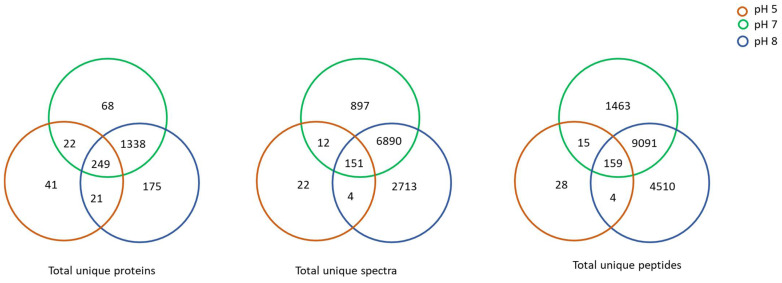
Venn diagrams summarizing the total unique proteins, spectra and peptides obtained from *B. subtilis* (EB2004S) CFS produced at pH 5, pH 7 and pH 8.

**Figure 5 ijms-23-15144-f005:**
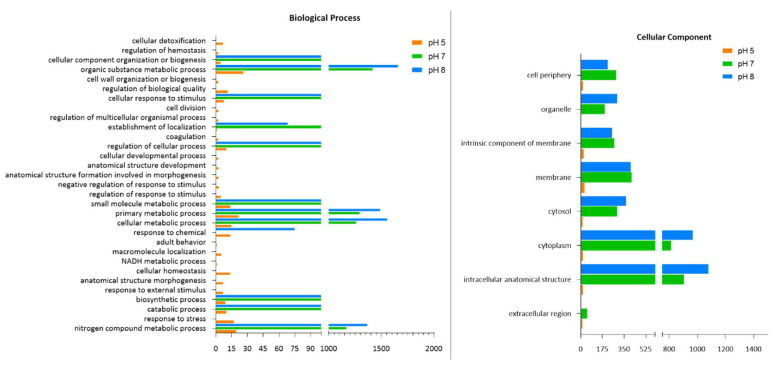
Variation in the biological processes, cellular components, molecular function and enzyme code distribution of proteins based on the GO enrichment analysis of *B. subtilis* obtained from CFS when cultured at pH 5, 7 and 8.

**Figure 6 ijms-23-15144-f006:**
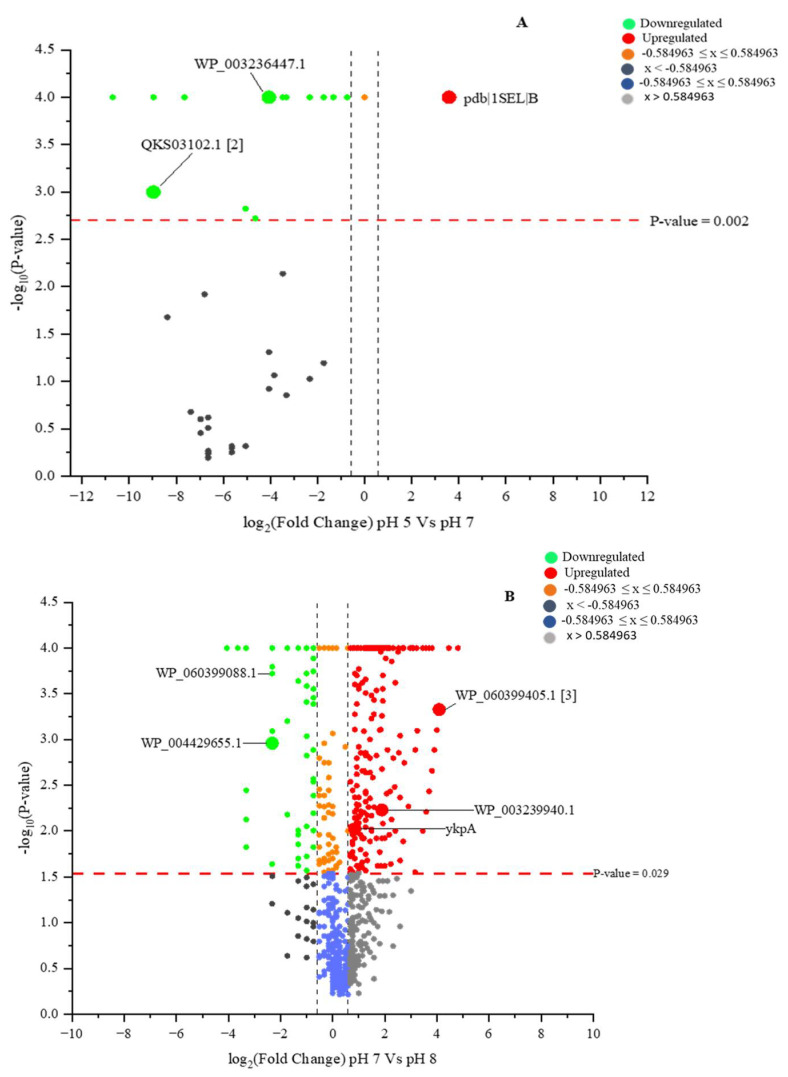
Volcano plots illustrating the distribution of differentially quantified proteins of *B. subtilis* CFS as −log10 (Benjamini–Hochberg-adjusted *p*-values) plotted against log2 (fold change) for (**A**): pH 7 vs. pH 5 and (**B**): pH 7 vs. pH 8. The non-axial black dashed vertical lines represent a ±1.5-fold change, while the horizontal red dashed line indicates the significance threshold (before logarithmic transformation) for (**A**): *p* ≤ 0.002 and (**B**): *p* ≤ 0.013.

**Table 1 ijms-23-15144-t001:** Summary of general characteristics of the bacterial species closely related to the studied strains (*B. subtilis* (EB2004S) and *L. helveticus* (EL2006H)) in previous studies.

Parameter	*B. subtilis*	*L. helveticus*
Genome size (bp)	4,214,630 [23,32]	21,000 [30]
Optimum growth pH range	7.0 [18,29]	5.5 to 5.8 [18,30]
Proteins protein-coding genes	4610 [33]	2462 [34]

**Table 2 ijms-23-15144-t002:** Differences in the enzyme code distribution of *B. subtilis* (EB2004S) and *L. helveticus* (EL2006H) CFS in comparison with the reference pH 7. The percentages in the parenthesis indicate increases (↑) and decreases (↓) in the expression of a given enzyme compared to the reference pH 7.

Enzyme Code Classes	*B. subtilis* (EB2004S)			*L. helveticus* (EL2006H)	
	pH 7	pH 5	pH 8	pH 7	pH 5
Hydrolases	591	34 (↓94%)	644 (↑9%)	431	118 (↓73%)
Isomerases	92	2 (↓98%)	123 (↑33%)	66.3	13.7 (↓79%)
Ligases	155	0 (↓100%)	188 (↑21%)	103.3	1.7 (↓98%)
Lyases	134	0 (↓100%)	199 (↑49%)	44.7	6.7 (↓85%)
Oxidoreductases	392	20 (↓95%)	490 (↑25%)	72	0 (↓100%)
Transferases	457	6 (↓98 %)	584 (↑28%)	194.7	3 (↓99%)
Translocases	95	0 (↓100%)	119 (↑26%)	31.3	6.7 (↓79%)

## Data Availability

Data supporting the reported results and conclusions will be made available by the authors, without undue reservation, together with those found in the online repositories of *MassIVE* under the dataset identifiers PXD036602 or doi:10.25345/C5XD0R27D for *Bacillus subtilis* and PXD036598 or doi:10.25345/C5PV6BB9K for *Lactobacillus helveticus*.

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
