# Peer review of "Secretome Analysis of the Plant Biostimulant Bacteria Strains Bacillus subtilis (EB2004S) and Lactobacillus helveticus (EL2006H) in Response to pH Changes"

_ijms, 2022, doi:10.3390/ijms232315144_

Round 1

Reviewer 1 Report

Summary

The paper “Secretome Analysis of Plant Biostimulant Bacteria Strains Bacillus subtilis (EB2004S) and Lactobacillus helveticus (EL2006H) in Response to pH Changes” is a continuation to the previous work of Msimbira et al., 2022 in which the plant growth promoting effects of supernatant obtained from the two strains at pH 5, 7 and 8 was investigated. In this previous study it was shown that supernatant of bacteria intubated at e.g. pH 5, has beneficial effects on tomato and corn plants incubated at pH 5. In this current study the secretome profiles of the supernatant of bacteria cultured at pH 5, 7 and 8 were analyzed. The authors showed that the at pH 5 most proteins were downregulated and a few unique proteins were found. The further investigation of this supernatant and its composition is important and interesting. Also, it should be explored which compounds are responsible for the beneficial effects. 

Minor Revisions:

Multiple parts of figure 2 were cut off, for example the y-axis and the legend. 

The peptides and spectra of the Venn diagrams (figure 1 and 4) were never discussed or explained in the text.  

Some proteins being detected would be associated with cell lysis. This should be discussed, even though this is irrelevant for the objective of this study with regard to the preliminary investigation. 

On page 2 line 86 it is said that protein expression was analyzed in log phase but later on page 11 lane 391 it is said that late exponential growth phase cultures (48 h) were used. Those statements can’t both be true. 

To assess the credibility and the accuracy of the results better, it should be stated, whether or not the experiments (sampling and analysis) were independently repeated and how many times. 

The volcano plots (figure 3 and 6) have multiple problems in their current state. What is the percentage of proteins determined to be significant outliers? Especially in figure 3 and 6 A there seem to be more outliers than usable datapoints resulting in plots without the typical volcano shape. In Figure 6, A and B are not marked in the figure and in A there is a datapoint highlighted without explanation. The information gained from the volcano plots is not discussed, explained or mentioned. The datapoints could be colored in four colors corresponding to the functional clusters established before. Also, proteins that have a high change in quantity paired with a good p-value should be highlighted and named in the figure. 

Author Response

Hello!

Find the attached response for the comments raised during review.

Thank you

Reviewer 2 Report

In this manuscript the authors identify the secretome of  Bacillus subtilis and Lactobacillus helveticus, grown at different pH. This work builds on previous  work in which the same authors showed that  CFS from these two strains showed positive effects on seed germination and seedling growth in corn and tomate (Msimbira et al., 2022). The finds could be important in the future if specific proteins from the secretome would be purified and related to  the probiotic action of these strains. However, from previous studies is not clear if a protein or other kind of molecule present in the CFS is involved in the plant promoting action.

I have also  other comments that the authors should consider:

1.             I can not find the supplementary file

2.             Is not clear if the proteomic analysis was done in the same bacterial growth condition (nutrient broth) in which the CFS were show to have an effect (Msimbira et al., 2022)

3.             Table 2, has a format problem (oxidoreductase row). The legend should refer the number in parenthesis.

4.             Line 267, should be sB and not dB

5.             Throughout the text the authors write about expression of proteins (line 159, 221…). Since the authors are looking for the secretome, is difficult to discriminate between expression or not being secreted, accumulating inside the cell. I think that this should be rephrase. It is known that some protein are only secreted as a response to changes in the pH.

6.         “The expression of proteins of unknown function are reported in various proteomic studies” (line 375-376). The same or others?

Author Response

Hello!

Find the attached response to the comments

Thank you

Reviewer 3 Report

This paper simply compared the secretome distribution of strains Bacillus subtilis and Lactobacillus helveticus at different pH conditions. These strains are very common rhizosphere growth promoters and have been fully commercialized. Therefore, there have been many studies on the stress resistance mechanism of these strains and their proteomics, and in fact, the revision of this paper has no great innovative significance.

The authors completed the experiment in the medium under different pH conditions, and obtained the proteomic enrichment of these strains under different pH conditions, but there was no further proof of the anti-disease or anti-stress function of these proteins.

In particular, the composition of proteins can be influenced by the interactions between microorganisms or between microorganisms and their environment. Therefore, whether the strains studied by the authors would have obtained these results in the presence of other indigenous microorganisms, assuming soil was used as a carrier, could not be replicated.

Author Response

Hello!

Find the attached responses to the comments

Thank you

Round 2

Reviewer 2 Report

Concerning point 5 I don't agree with the authors. Consequently in point 6 I would write identification and not expression.

Author Response

Hello!

Thank you for your time, the comments are attached.

Reviewer 3 Report

It will be very interesting to verify the function of these strains in soil

Author Response

Hello!

Thank you for your time and comments.

The responses are attached here
